# HIF-1α and Pro-Inflammatory Signaling Improves the Immunomodulatory Activity of MSC-Derived Extracellular Vesicles

**DOI:** 10.3390/ijms22073416

**Published:** 2021-03-26

**Authors:** Marta Gómez-Ferrer, Estela Villanueva-Badenas, Rafael Sánchez-Sánchez, Christian M. Sánchez-López, Maria Carmen Baquero, Pilar Sepúlveda, Akaitz Dorronsoro

**Affiliations:** 1Regenerative Medicine and Heart Transplantation Unit, Instituto de Investigación Sanitaria La Fe, 46026 Valencia, Spain; margofe22@gmail.com (M.G.-F.); estela.villanueva.badenas@gmail.com (E.V.-B.); rafa_4586@hotmail.com (R.S.-S.); 2Àrea de Parasitologia, Departament de Farmàcia i Tecnologia Farmacèutica i Parasitologia, Universitat de València, Av. V.A. Estellés s/n, 46100 Valencia, Spain; chsanlo@alumni.uv.es; 3Joint Research Unit on Endocrinology, Nutrition and Clinical Dietetics, Instituto de Investigación Sanitaria La Fe, Universitat de Valencia, 46026 Valencia, Spain; 4Servicio de Cirugía Oral y Maxilofacial, Hospital Universitari I Politècnic La Fe, Avenida, Fernando Abril Martorell, 106, 46026 Valencia, Spain; baquero_car@gva.es

**Keywords:** mesenchymal stromal cells, T-cells, extracellular vesicles, hypoxia-inducible factor 1-alpha, immunomodulation

## Abstract

Despite the strong evidence for the immunomodulatory activity of mesenchymal stromal cells (MSCs), clinical trials have so far failed to clearly show benefit, likely reflecting methodological shortcomings and lack of standardization. MSC-mediated tissue repair is commonly believed to occur in a paracrine manner, and it has been stated that extracellular vesicles (EVs) secreted by MSCs (EVMSC) are able to recapitulate the immunosuppressive properties of parental cells. As a next step, clinical trials to corroborate preclinical studies should be performed. However, effective dose in large mammals, including humans, is quite high and EVs industrial production is hindered by the proliferative senescence that affects MSCs during massive cell expansion. We generated a genetically modified MSC cell line overexpressing hypoxia-inducible factor 1-alpha and telomerase to increase the therapeutic potency of EVMSC and facilitate their large-scale production. We also developed a cytokine-based preconditioning culture medium to prime the immunomodulatory response of secreted EVs (EV_MSC-T-HIF_^c^). We tested the efficacy of this system in vitro and in a delayed-type hypersensitivity mouse model. MSC-T with an HIF-1α-GFP lentiviral vector (MSC-T-HIF) can be effectively expanded to obtain large amounts of EVs without major changes in cell phenotype and EVs composition. EV_MSC-T-HIF_^c^ suppressed the proliferation of activated T-cells more effectively than did EVs from unmodified MSC in vitro, and significantly blunted the ear-swelling response in vivo by inhibiting cell infiltration and improving tissue integrity. We have developed a long-lived EV source that secretes high quantities of immunosuppressive EVs, facilitating a more standard and cost-effective therapeutic product.

## 1. Introduction

Mesenchymal stromal cells (MSCs) are multipotent progenitor cells that have the ability to differentiate into chondrocyte, osteoblast and adipocyte lineages [1]. MSCs have recently emerged as potential therapeutic agents to treat multiple diseases based both on their tissue regeneration and their immunosuppression capacity, and numerous preclinical studies have demonstrated that MSCs can ameliorate tissue damage after ischemic events and modulate acute and chronic inflammatory reactions [2]. Mechanistically, the reparative capacity of MSCs is generally linked to their paracrine effects on target tissue/cells, and in particular the production of soluble factors, which are involved in the healing process. In terms of immunosuppression, MSCs are able to inhibit the proliferation of activated T-cells in a dose-dependent manner mainly through their expression of indoleamine 2,3 dioxygenase (IDO) and cyclooxygenase 2 (COX2) [3,4]. In addition, MSCs induce the production of regulatory T-cells by secreting transforming growth factor-β1 [5]. Likewise, MSCs are able to inhibit the activation of pro-inflammatory M1 macrophages and promote M1 to M2 polarization [6,7]. Finally, MSCs can suppress the proliferation and cytotoxic activity of natural killer (NK) T-cells [4,8,9].

The properties mentioned above, combined with the intrinsic homing capacity of MSCs to injured tissues, make them excellent candidates for cell-based therapeutic strategies to treat a wide range of disorders. However, almost all clinical trials performed thus far have been inconsistent and have failed to demonstrate unequivocal efficacy. Indeed, only one trial designed by the cell therapy company Tigenix (in 2018) for the treatment of refractory rectovaginal anal fistula in the setting of Crohn’s disease achieved significant positive results. A positive aspect of the clinical trials, however, is that no adverse events were reported directly related to MSC infusion, supporting further research to develop new therapeutic strategies for bench to bedside translation [10].

Extracellular vesicles (EVs) are small particles 30–200 nm in diameter delimited by a lipid bilayer membrane that is actively released by cells. EVs have a key role in intercellular communication and are capable of carrying a diverse cargo of different cytokines, membrane trafficking molecules, chemokines, heat shock proteins and even mRNAs and microRNAs [11,12]. Several preclinical and clinical studies have shown that EVs can partially recapitulate the therapeutic effects of whole cells in the context of transplantation. In fact, several studies reported that EVs secreted by MSCs modulate the activity of different immune cells both in vitro and in vivo [13,14]. Despite these encouraging findings, several challenges remain for MSC-derived EVs to be clinically useful. For instance, the effective dose used in animal models is very high and the observed effects are modest. Furthermore, MSCs are primary cells that ultimately senesce after a limited number of divisions, obliging the replacement of the parental cells and hindering the standardization and manufacturing processes.

Hypoxia-inducible factor 1-alpha (HIF-1α) is a master regulator of the cellular adaptation to hypoxia that modulates myriad processes in MSCs such as survival, proliferation, migration and differentiation [15,16]. Several studies have demonstrated significant improvements in the regeneration potential of MSCs after hypoxia preconditioning [17,18]. We previously established that the overexpression of HIF-1α in MSCs potentiates their immunosuppressive capacity on different immune cell populations including dendritic cells, NK cells and monocytes [19]. In this context, it is important to consider that the immunosuppressive ability of MSCs is not constitutive, but is instead induced by crosstalk with different cells of the immune system. This is clearly important for any EV-based therapy since the healing potential of the released EVs will depend on the physiological state of the parental cells. Accordingly, the immunosuppressive properties of MSC must be triggered or primed by external stimuli prior to EV collection in order to transfer the immunomodulatory capacity to EVs before they are released [20].

In the present study, we used a combination of genetic modification and pro-inflammatory stimulation on MSCs to circumvent some of the limitations of MSC-based therapy for tissue repair. In doing so, we have been able to improve the therapeutic capacity of EVs and to obtain a consistent and virtually unlimited source of highly immunosuppressive EVs.

## 2. Results

### 2.1. Overexpression of hTERT in Dental Pulp-Derived MSCs Generates a Non-Senescent Cell Line

A major challenge in the development of MSC-based therapies is the generation of a sufficiently large number of cells for transplantation. To address this issue, we immortalized dental pulp-derived MSC primary cultures by overexpressing human telomerase enzyme (MSC-T) using a lentiviral vector (Appendix A). Transgene integration provided resistance to hygromycin (Appendix A) and increased telomerase expression and activity (Appendix A) in MSCs. Consequently, the MSC-T cultures overcame proliferative exhaustion and were able to maintain division long after the unmodified MSCs reached replicative senescence (Appendix A). With this approach, we significantly extended the time in which an MSC cell line can be used as a stable source of EVs.

### 2.2. Activation of MSC-T Triggers Expression of Immunosuppressive Molecules and Increases EV Secretion

Steady-state MSCs have a low immunosuppressive capacity and require stimulation with pro-inflammatory molecules. Based on previous works, we developed a conditioning medium composed of TNF-α, IFN-γ and IL-1-β that mimics a pro-inflammatory environment. As shown in Figure 1, cytokine-activated MSC-T (MSC-T^c^) overexpressed several genes related to immunosuppression, such as *IDO*, *COX2* and *PD-L1* (Figure 1a), which was corroborated by Western blotting (Figure 1b) and flow cytometry (Figure 1c). Remarkably, we observed that compared with unmodified MSC, MSC-T^c^ released a significantly higher amount of EVs, as measured by bicinchoninic acid (BCA) and acetylcholinesterase assays and by nanoparticle tracking analysis (NTA) (Figure 1d–f) without alterations to other phenotypic features such as morphology, size and presence of EV markers (Figure 1g–i).

### 2.3. Conditioning of MSC-T Increases the Immunosuppressive Capacity of EVs

Based on the higher expression of immunomodulatory molecules observed in MSC-T^c^, we next evaluated their capacity to secrete immunosuppressive vesicles. We assessed the immunomodulation capacity of EVs secreted by MSC-T and MSC-T^c^ using a T-cell proliferation assay and measured CFSE dilution of stained T-cells stimulated with CD3/CD28 beads. As shown in Figure 2, MSC-T^c^-derived EVs (EV_MSC-T_^c^) regulated T-cell proliferation more effectively than did EV_MSC-T_ with both CD4 and CD8 cells (Figure 2a, Appendix A). To better understand the mechanism behind this phenomenon, we measured the levels of different immunosuppressive molecules and found that EV_MSC-T_^c^ were enriched for IDO, COX2 and PD-L1 in comparison with EV_MSC-T_ (Figure 2b). While IDO and COX2 exert their suppressive effects as intermediate players of a chain of reactions, the PD-L1/PD-1 signaling pathway is initiated when both molecules come into contact at the immunological synapse [21]. In this regard, we observed that PD-L1 was not only present in the membrane of EVs poised to interact with PD-1 in target cells, but it was also significantly upregulated when parental cells were activated by pro-inflammatory molecules (Figure 2c), suggesting a relevant role of this signaling pathway in the observed immunosuppression.

### 2.4. HIF-1α Overexpression in MSC-T Increases the Immunomodulatory Capacity of Secreted EVs

We previously showed that MSCs overexpressing HIF-1α have an enhanced immunosuppressive capacity [19]. Thus, we sought to evaluate whether the immunomodulatory benefits triggered by HIF-1α overexpression were conserved in secreted EVs. Thus, we transduced MSC-T with an HIF-1α-GFP lentiviral vector (MSC-T-HIF) or GFP lentiviral vector (MSC-T) to generate new cell lines to be used as the EV source. Transduction efficiency was measured by flow cytometry, which detected GFP in transduced cells, and the abundance of the transgene was quantified by Western blotting and qPCR. MSC-T-HIF showed similar features to MSC-T in terms of cell life-span and other MSC characteristic markers (Appendix A).

The NF-kB pathway is a key regulator of the immunosuppressive capacity of MSCs. It has been stated that activation of NF-κB in MSC is essential to stimulate the modulatory capacity of stromal cells [22]. Thus, we measured NF-κB pathway activity of both MSC-T and MSC-T-HIF after conditioning with the cytokine cocktail by measuring the nuclear translocation of p65 and the expression levels of IL-6 and other immunosuppressive molecules. We observed that overexpression of HIF-1α triggered a stronger p65 nuclear translocation after exposing MSCs to the conditioning cocktail (Figure 3a). Furthermore, conditioned MSC-T-HIF (MSC-T-HIF^c^) expressed higher levels of IL-6 and other immunosuppressive molecules, such as IDO, COX2 and PD-L1 (Figure 3b,c).

In terms of EV secretion, we observed that MSC-T-HIF^c^ secreted higher amounts of EVs than did MSC-T^c^ without significant alterations in their characteristic features such as size, shape or tetraspanins content (Figure 4a–f), in line with our previously published results [23,24]. Next, we measured the immunosuppressive capacity of MSC-T-HIF^c^-derived EVs (EV_MSC-T-HIF_^c^) in a T-cell proliferation assay. As shown in Figure 4g, EV_MSC-T-HIF_^c^ had a stronger capacity to suppress T-cell proliferation than EV_MSC-T_^c^ counterparts, affecting both CD4 and CD8 subpopulations.

To gain more insight into the mechanisms that triggered the immunosuppressive features of EV_MSC-T-HIF_^c^, we analyzed the levels of IDO and PDL-1 by Western blot and found that levels were higher in EV_MSC-T-HIF_^c^ than in EV_MSC-T_^c^ (Figure 4h). The increased presence of PDL-1 on the surface of EVs was corroborated by flow cytometry (Figure 4i). Regarding the expression levels of COX-2 by EV_MSC-T-HIFc_ compared to EV_MSC-Tc_ we obtained inconsistent results (data not shown).

### 2.5. EV_MSC-T-HIF_^c^ Have Improved Immunoregulatory Potential in a Delayed-Type Hypersensitivity Mouse Model

In vitro assays used in immunology research are useful to study the role of treatments on each cell type; however, in vivo immune reactions are coordinated multicellular processes and require animal models to unravel their complexity. Thus, we tested the immunosuppressive capacity of EV_MSC-T-HIF_^c^ in a delayed-type hypersensitivity (DTH) mouse model, as a suitable approach for evaluating the immune response mediated by T-cells and monocytes/macrophages [25]. Using this model, we observed that EV_MSC-T-HIF_^c^ were significantly more effective than EV_MSC-T_^c^ and the control (phosphate buffered saline) to reduce ear swelling (Figure 5a). As shown in Figure 5b, EV_MSC-T-HIF_^c^ reduced swelling to 38.64 ± 2.88% of control values (non-MSC-treated) 48 h after challenge and to 40.76 ± 7.18% at 72 h. In addition to swelling, DTH is associated with epidermal hyperplasia and CD45+ cell infiltration. Histological analysis of ear preparations showed that EV_MSC-T-HIF_^c^ reduced hyperplasia and cell infiltration substantially when compared with EV_MSC-T_^c^ and control treatments (Figure 5c,d).

Cell infiltration was further characterized by quantifying CD45+ cell infiltration in the challenged ear (Figure 6a) showing that leucocyte infiltration is dramatically reduced in mice treated with a single dose of EVs. DTH inflammatory reaction is driven by T-cell–monocyte interaction, monocyte polarization being a relevant regulator [26]. Therefore, we analyzed the presence of M1 (pro-inflammatory) and M2 (immunomodulatory) [26] macrophages in the ears (Figure 6b,c). We observed that at 72 h time point, ears not treated with EVs showed a high M1 infiltrate while the presence of M2 macrophages was scarce. On the other hand, EV_MSC-T_^c^ transplanted mice showed less M1 infiltrate and higher presence of M2 cells in the ears. Moreover, when EV_MSC-T-HIF_^c^ are injected, the polarization to macrophages towards an immunosuppressive profile is exacerbated.

## 3. Discussion

Due to the lack of unequivocally positive results obtained in the many clinical trials performed to date [10], researchers have gone back to the bench to design more efficient MSC-based therapeutic strategies. In this regard, EVs secreted by stromal cells might be a good candidate as a next-generation therapy. In the present study, we confirm previous reports [13,27,28] that MSC-derived EVs can reproduce the immunosuppressive features of the parental cells and modulate the proliferation of both CD4^+^ and CD8^+^ subpopulations. In addition, we observed that MSC-derived EVs are loaded with multiple molecules with immunomodulation capacity, such as IDO and PD-L1.

The use of EVs as a therapy has significant advantages over MSCs in terms of safety, manufacturing and logistics. For instance, because of their small diameter, EVs can be sterilized by filtration. Accordingly, clean room standards might not have to be as high as those necessary for advanced therapy medicinal product production, and the use of high clean room standards would be required only to sterilize the final product. At the same time, EVs are much easier to manage than cells. Freezing, thawing and storage conditions are less critical for the integrity of EVs than for cells, which should make bedside manipulation easier. Furthermore, EVs are safer to use in patients than live cells. Vesicles have no nuclear DNA and are non-replicative elements, allaying the concerns of malignant transformation after infusion [29,30]. Due to minimal risk of transgene transfer after transplantation, it is possible to manipulate EV-secreting cells both genetically and by signaling factors in order to boost the therapeutic features of the obtained products.

MSCs do not divide unlimitedly and lose therapeutic features with extended periods in culture. Thus, it is extremely challenging to adequately design and standardize any therapy based on EVs secreted by MSCs. Accordingly, we elected to genetically modify EV-secreting cells (by hTERT and HIF-1α overexpression) to overcome this limitation, as immortalized cell lines will likely be essential to facilitate future manufacturing processes. Genetic modification of MSCs has been used previously as a tool to investigate biological behavior and potential therapeutic mechanisms, but it never was intended to be used for therapy in humans due to safety concerns [31]. However, using EVs as final products overcomes this limitation since they lack replicative genetic material that could harm patients. In this manner, genetic tools can be used on secreting cells to improve their therapeutic features, potentially heralding a new era in cell therapy.

hTERT immortalization is a well-described modification of MSCs to circumvent senescence and favor more time in culture without altering therapeutic features [32,33,34]. Indeed, we observed that hTERT-MSC had increased lifespan in culture without losing the capacity to secrete immunosuppressive EVs. Previously Chen et al. immortalized hESC-MSCs by overexpressing *c-myc* in order to obtain a consistent source of EVs [35]. However, using an oncogene significantly alters the biological features of cells [36] and although the authors showed that the therapeutic capacity of EVs was maintained and no presence of c-myc protein was detected, this strategy raises safety concerns. By contrast, our data and those of others indicate that hTERT will be safer since it does not profoundly modify MSC behavior and even improves some relevant features of MSC [37,38,39].

Several studies have reported that growing MSCs in hypoxia enhances their therapeutic potential [17,18,23,40] as well as that of EVs. However, maintaining cells in low oxygen is difficult and exposure to a normoxic environment rapidly reverses the gained features. We previously observed that overexpression of HIF-1α increases the immunosuppressive capacity of MSC [19] and that this genetic modification was able to potentiate the angiogenic potential of EVMSC-HIF [23]. In this manuscript, we wanted to go a step further and evaluate if overexpression of HIF-1α together with MSC licensing with pro-inflammatory factors could also improve immunomodulatory properties of secreted EVs.

The improved immunomodulatory properties of EV_MSC-T-HIF_^c^ were supported by the following observations: (i) they were able to induce immunosuppression in both CD4 and CD8 cell populations, (ii) they expressed the immunomodulatory factors PDL1 and IDO, (iii) they reduced leukocyte infiltration and induced M2 polarization in oxazolone-treated animals.

Thus, our study opens the door for future studies to evaluate whether the genetic modification protocols used previously to better understand the biology of cell therapy can now be reframed as translational studies and used as stepping stones for new therapeutic strategies.

EVs derived from MSCs cultured in standard conditions have immunosuppressive properties [13,14]; however, their potency is limited and so the effective dose of EVs needed is very high, which makes developing a therapy challenging. For instance, Kordelas et al. treated a patient with graft versus host disease using a dose of EVs equivalent to the secretion from ~300 million MSC [14]. In this study, clinical GvHD symptoms improved significantly shortly after the start of the therapy, but no mid-term or total remission was seen. We used a priming cocktail comprising IL-1β, TNF-α and IFN-γ to increase the potency of EVs secreted by MSCs and achieve an effective treatment. This cocktail was based on previously published data aiming to activate MSCs [20,41]. Among the pro-inflammatory cytokines, TNF-α, IL-1β and IFN-γ are the most important molecules related to immunosuppressive activation of MSCs, with each driving different elements. TNF-α activates the NF-κB pathway, which plays a central role in the regulation of immunosuppression by MSCs [22]. IL-1β upregulates migration and homing molecules on the surface of MSC by signaling through NF-κB pathway [42] and also primes other immunomodulatory molecules [43]. Finally, IFN-γ triggers IDO expression, which is responsible for suppressing the activation and proliferation of multiple immune cells by depleting tryptophan and producing kynurenine [44].

We found that stimulation/priming of parental MSCs increased the loading of immunosuppressive molecules in EVs and also increased the immunosuppression capacity of EVs in vitro almost 3-fold, affecting both CD4 and CD8 subpopulations. These results are in line with a report by Kim et al., who showed that EVs derived from IFN-γ-primed MSCs exerted immunosuppressive effects in a GvHD murine model [45]. Our results indicate that priming MSCs is likely essential for boosting their immunomodulatory effects.

While further preclinical and clinical studies will be necessary to investigate the mechanisms underlying the seemingly excellent therapeutic capacity displayed by EV_MSC-T-HIF_^c^, we believe that our strategy holds great promise for practical implementation.

## 4. Materials and Methods

### 4.1. Ethical Statements

All subjects gave their informed consent for inclusion before they participated in the study. The study was conducted in accordance with the Declaration of Helsinki, and the protocol was approved by the Ethics Committee of The Hospital La Fe Universitari i Politècnic (Project identification 2019/0101).

Animal procedures were approved by Hospital La Fe Ethics Committee (Protocol N° 2018/VSC/PEA/0149) according to guidelines from Directive 2010/63/EU of the European Parliament on the protection of animals used for scientific purposes.

### 4.2. Human Samples

Human dental pulp samples were obtained from third molars, which were extracted for orthodontic reasons from healthy young people (18–21 years of age) who gave their informed consent or directly from Bank (Inbiobank, Guipuzcoa, Spain).

Buffy coats of healthy donors were obtained from Centro de Transfusión de la Comunidad Valencia (Valencia, Spain) after informed consent.

### 4.3. Cell Culture

Human dental pulp MSCs were cultured as previously described [46]. Briefly, pulp tissue was minced into small fragments (<1 mm^3^) prior to digestion in a solution of 2 mg/mL collagenase type I (Gibco, Grand Islands, NY, USA) for 90 min at 37 °C and primary cultures were established. Cells were cultured in Dulbecco’s Modified Eagle Medium (DMEM) low glucose (Sigma-Aldrich) supplemented with 10% heat-inactivated fetal bovine serum (FBS; Corning), 100 U/mL penicillin and 100 μg/mL streptomycin (P/S, Millipore). To obtain EVs, MSCs were cultured in extraction medium (EM), which was prepared by supplementing DMEM with 10% EV-depleted FBS and antibiotics. EV-depleted FBS was generated by ultracentrifugation of regular FBS and DMEM mixed 1:1 at 100,000× *g* for 16 h. Buffy coats of healthy donors were obtained from the Centro de Transfusión de la Comunidad Valencia, Spain after informed consent. PBMCs were isolated by density gradient centrifugation with Histopaque (Sigma-Aldrich, Darmstadt, Germany). Isolated PBMCs were cultured in RPMI (Thermo-Fisher Scientific, Waltham, MA, USA) supplemented with 10% FBS, 2 mM glutamine, 100 U/mL penicillin and 100 μg/mL streptomycin (P/S, Millipore, Burlington, MA, USA).

### 4.4. MSC Conditioning Medium

MSCs were primed by incubation in a medium consisting of EM supplemented with IFN-γ (50 ng/mL) (R&D), TNFα (10 ng/mL) (R&D) and IL-1β (10 ng/mL) (Promega, Madison, WI, USA).

### 4.5. Lentiviral Production and MSC Transduction

Viral particles were produced in human embryonic kidney 293T cells (ATCC^®^ CRL-3216™ www.atcc.org accessed on 22 March 2021). Briefly, 293T cells were seeded in high-glucose DMEM containing 10% FBS. pMD2.G (Addgene, 12259), psPAX2 (Addgene, 12260) and the lentiviral vector carrying the transgene were transfected into the packaging cell line by calcium phosphate precipitation. MSC transduction was carried out at a multiplicity of infection of 10 in order to achieve >95% infection. pLV-hTERT-IRES-hygro was obtained from Addgene (Addgene, 85140). The efficiency of transduction was evaluated by a hygromycin selection test. The pWPI-HIF-1α-GFP and pWPI-GFP vectors have previously been described [18]. Transduction efficiency was evaluated by quantifying GFP+ cells by flow cytometry. MSC-T cells were transduced with pWPI-GFP as a control (named MSC-T). GFP-positive cells were sorted to obtain pure cultures. Population doubling times (PDT) were calculated using the formula PDT = (t2 − t1)/3.32 × (logN2 − logN1) where N2 is the cell harvest number and N1 the inoculum cell number [46].

### 4.6. Hygromycin Selection Test

Different MSC lines were seeded at 2.5 × 10^4^ cells/cm^2^ and incubated with 100 µg/mL of hygromycin for 48 h. Cell viability was measured using the MTT reduction assay. Briefly, after incubation, cells were washed and 0.5 mg/mL MTT solution was added for 3 h. Intracellular formazan was released with DMSO and absorbance was measured by absorbance at 550 nm in a Halo Led 96 spectrophotometer (Dynamica Scientific Ltd., Livingston, United Kingdom).

### 4.7. Telomerase Activity

Telomerase activity was determined using a protocol described by Erbert et al. [47]. Briefly, cell lysates were incubated with TS primer (5’AATCCGTCGAGCAGAGTT, Sigma-Aldrich), ACX reverse primer (5’GCGCGGCTTACCCTTACCCTTACCCTAACC, Sigma-Aldrich), SYBR Green PCR Master Mix (Applied Biosystems, Foster City, California, United States) and at 25 °C for 20 min. Reaction products were quantified in a ViiA 7 Real-time PCR system. Relative telomerase activity (RTA) was extrapolated from a standard curve performed with HEK-293T cells.

### 4.8. Senescence-Associated β-Galactosidase Assay

A senescence-associated β-galactosidase assay was performed on MSCs seeded at low density using a β-galactosidase staining kit (#9860, CST).

### 4.9. EV Isolation and Characterization

EVs were isolated using a serial ultracentrifugation protocol [24]. Briefly, EV-containing supernatants were centrifuged at 2000× *g* for 20 min (Eppendorf 5804 benchtop centrifuge, A-4-62 rotor), followed by centrifugation at 10,000× *g* for 70 min (Hitachi CP100NX centrifuge, Beckman Coulter 50.2 Ti rotor) and supernatants were subsequently filtered manually through a 0.22 μm using a syringe. EVs were then concentrated by two rounds of ultracentrifugation at 110,000× *g* for 120 min (Hitachi CP100NX centrifuge, Beckman Coulter 50.2 Ti rotor) with a washing step in between rounds. When required, samples were passed through a 100 kDa Amicon^TM^ filter (Merck Millipore, Burlington, MA, USA) to remove the added cytokines and were then filtered through a 0.22 μm filter to maintain sterility. EV protein concentration was determined with the Pierce BCA Protein Assay Kit (Thermo Fisher Scientific) to ensure that equal amounts of protein were used for experiments. EVs were suspended in RIPA buffer (1% NP40, 0.5% deoxycholate, 0.1% sodium dodecyl sulfate in Tris-buffered saline (TBS), (Sigma-Aldrich)) for Western blotting and in PBS for characterization and functional analysis. EV size distribution and quantification of vesicles were analyzed by NTA using a NanoSight NS3000 System (Malvern Instruments, Malvern, United Kingdom). Electron microscopy was performed as described [48]. Briefly, EVs were diluted in PBS, loaded onto Formwar carbon-coated grids, contrasted with 2% uranyl acetate and finally examined with a FEI Tecnai G2 Spirit transmission electron microscope. Images were acquired using a Morada CCD Camera (Olympus Soft Image Solutions GmbH, Münster, Germany). An acetylcholinesterase assay was developed as previously described [23]. Thirty microliters of each EV preparation were added to individual wells of a 96-well flat-bottomed microplate. Then, 1.25 mM acetylthiocholine and 0.1 mM 5,5’ dithiobis (2-nitrobenzoic acid) (Sigma-Aldrich, Darmstadt, Germany) were added to each well to a final volume of 300 μL and the change in absorbance at 412 nm was monitored every 10 min. Data were represented as acetylcholinesterase activity after 30 min of incubation at 37 °C and MSC-T set as 1 to facilitate comparison.

### 4.10. T-cell Proliferation Assays

A T-cell proliferation assay was performed as described [22]. Prior to culturing, PBMCs were labeled with 5 μM carboxyfluoroscein succinimidyl ester (CFSE; Thermo Fisher Scientific) and activated with Dynabeads™ Human T-Activator CD3/CD28 (Thermo Fisher Scientific). To evaluate the immunosuppressive potential, we added 15 µg of different EV extracts to 10^5^ CFSE-labeled PBMCs in 0.5 mL of medium. After 5 days of culture, proliferation of T-cells was determined by flow cytometry to measure CFSE dilution. Analyses of flow cytometry data and the expansion index (EI) [49], were performed using FlowJo software (FlowJo LLC, BD, Franklin Lakes, New Jersey, United States). Percentage of immunosuppression was calculated normalizing data to a 0–100% scale by establishing 0% immunosuppression for the EI of activated PBMCs not treated with EVs (Ac) and 100% immunosuppression for non-activated PBMCs’ (no Ac) EI using the following formula $\%\;Immunosuppression=\left(EI_{Ac}\;-\;EI_{EVs}\right)\;/\;\left(EI_{Ac}\;-\;EI_{none\;Ac}\right)\;\times 100$ CD3^+^ T-cells and CD4^+^ and CD8^+^ positive subpopulations were identified by cell surface antigen staining.

### 4.11. Immunofluorescence

To determine the intracellular localization of p65, cells were fixed and permeabilized with cold 70% ethanol solution and staining was performed using a goat anti-human p65 (Santa Cruz, sc-372) antibody and a goat anti-rabbit IgG (1:1000; Alexa Fluor^®^ 568, Abcam, Cambridge, United Kingdom) secondary antibody. Nuclei were stained with DAPI. Quantification of fluorescence location was performed using ImageJ (NIH). Each point shown in the graph corresponds to the mean of 50 cells.

### 4.12. Real Time Quantitative PCR

RNA was extracted using TRIzol reagent (Invitrogen) and purified with the RNeasy Plus Mini Kit (Qiagen, Hilden, Germany). cDNA was obtained by reverse transcription using M-MLV reverse transcriptase (Invitrogen, Carlsbad, California, United States). RT-qPCR was performed with SYBR™ Green PCR Master Mix (Applied Biosystems) and the following human-specific sense and antisense primers were designed using: *GAPDH* CCCCTCTGCTGATGCCCCA (F) and TGACCTTGGCCAGGGGTGCT (R); *TERT* GTATGGCTGCGTGGTGAACT (F) and TCTGAACAAAAGCCGTGCCA (R); *IDO* CTACCATCTGCAAATCGTG (F) and GAAGGGTCTTCAGAGGTCT; *COX2* GAATCATTCACCAGGCAAA (F) and TCTGTACTGCGGGTGGAACA (*R*)*; PD-L1 CCTGCAGGGCATTCCAGAAA (F) and GTCCTTGGGAACCGTGACAG (R); IL6* CCAGGAGCCCAGCTATGAAC (F) and GAGCAGCCCCAGGGAGAA (R); *HIF-1α* CACAGCCTGGATATGAA (F) and GAATTCTTGGTGTTATATATG (R). The reaction was performed using a Viia 7 PCR System (Applied Biosystems).

### 4.13. Western Blot Analysis

EVs or cells were lysed in 100 μL of RIPA buffer containing protease (Complete, Sigma-Aldrich) and phosphatase (PhosSTOP, Sigma-Aldrich) inhibitors. Equal amounts of protein samples were mixed with non-reducing Laemmli sample buffer (BioRad) and denatured at 96 °C for 5 min. Proteins were separated on 10% SDS-polyacrylamide gels and transferred to polyvinylidene difluoride membranes (Immobilon-P; Millipore). Membranes were blocked with TBS containing 5% (*w*/*v*) non-fat dry milk powder with 0.1% Tween-20. Human primary antibodies used for Western blotting were: anti-tubulin (dilution 1/4000; Sigma-Aldrich; T5168), anti-hTERT (dilution 1/500; Rockland Immunochemicals; 600-401-252S), anti-HIF-1α (dilution 1/500; BD Biosciences; 610958), anti-IDO (dilution 1/1000; Cell Signaling Technology; #12006), anti-COX2 (dilution 1/1000; Santa Cruz; H-3), anti-IL-6 (dilution 1/1000; Santa Cruz; H-183), anti-HSP70 (dilution 1/500; Cell Signaling Technology; D69), anti-CD9 (dilution 1/500; Santa Cruz; C-4), anti-Tsg101 (dilution 1/200; Santa Cruz; C-2), anti-calnexin (dilution 1/1000; Santa Cruz, H-70) and anti-PD-L1 (dilution 1/500; AB Clonal; A11273). Secondary antibodies used were anti-IgG rabbit (dilution 1/4000; Dako; P0448) and anti-IgG mouse (dilution 1/10,000; Sigma-Aldrich; A9044). Detection was carried out using peroxidase-conjugated secondary antibodies and the ECL Plus Reagent (GE Healthcare, Chicago, IL, USA). Reactions were visualized using an Amershan Imager 600 (GE Healthcare) and quantified with ImageJ software (NIH, Bethesda, MD, USA).

### 4.14. Flow Cytometry

For flow cytometry analysis, EVs or cells were first incubated with a blocking solution (PBS containing 1% normal mouse serum) for 10 min and then incubated with saturating amounts of fluorochrome-conjugated antibodies for 30 min at 4 °C. Human antibodies used were: anti-CD14 (RPE, Dako; TUK4), anti-CD34 (APC, Dako; C7238), anti-CD45 (PECy5, BD Biosciences; HI-30), anti-CD73 (BV421, BD Biosciences; AD2), anti-CD90 (PE, Immunotech; F15-42-1-5), anti-CD105 (PE, Abcam; SN6), anti-CD3 (PerCP-Cy, BD Biosciences; SK7), anti-CD4 (BV510, BD Biosciences; L200), anti-CD8 (PE-Cy7, BD Biosciences; RPA-T8) and anti-CD274 (APC, BD Biosciences; MIH1) at concentrations recommended by manufacturers, and were analyzed using a BD FACSCANTO II flow cytometer equipped with Flowjo^®^ software (FlowJo LLC, BD, Franklin Lakes, NJ, USA).

EVs were analyzed using a previously described capturing bead-based protocol [50]. Briefly, 10-μm streptavidin-coated magnetic beads (Millipore) were coupled with 1 μg of biotinylated antibodies (anti-CD81, anti-CD9 and anti-CD63, immunostep) for 1 h at room temperature, under gentle agitation. Beads were washed and incubated with 30 μg of EVs overnight at 4 °C with constant, gentle rotation. EV-coated beads were washed, blocked and stained as described above.

### 4.15. Delayed-Type Hypersensitivity Mouse Model

Adult male Balb/c mice (6 weeks old, 22–26 g) were purchased from Charles River Laboratories Inc. and housed at the IIS La Fe animal housing facility under standard conditions. Animal procedures were approved by institutional ethical and animal care local committees according to guidelines from Directive 2010/63/EU of the European Parliament on the protection of animals used for scientific purposes.

Mice were shaved and sensitized through abdominal topical application of 150 µL of 3% oxazolone (Santa Cruz) dissolved in acetone, 5 days before the experiment. On day 5 post-sensitization, the mice were challenged by topical application of 20 µL of 1% of oxazolone dissolved in acetone on the inner and outer left ear surfaces. After 6 h of challenge, 15 µg of EVs were administered by subcutaneous injection in the back of the ear using 50 µL of sterilized and filtered PBS as a vehicle. The control group was administered only with vehicle. Ear thickness was measured at pre-sensitization and at 6, 24, 48 and 72 h after challenge with a digital micrometer. The degree of swelling was calculated as the thickness difference between the left ear (challenged ear) and the right ear (unchallenged ear). The mice were then sacrificed, and the ears were collected for morphometric analysis.

### 4.16. Histology and Immunofluorescence

Ear samples were fixed with ethanol, embedded in paraffin, cut into 5-µm-thick sections and stained with hematoxylin–eosin (Sigma-Aldrich). Slides were visualized using a Leica DMD108 Digital Microscope (Leica Microsystems). For immunofluorescence analysis, slides were blocked with 5% neonatal goat serum and 0.1% Triton X-100 in PBS for 1 h. Slides were incubated with anti-CD45 (dilution 1/200; BD Biosciences), anti-F4/F80 (dilution 1/200; Abcam), anti-CD274 (dilution 1/200; AB Clonal A11273) or anti-CD206 (dilution 1/200; Abcam) overnight in a humidified chamber at 4 °C and were washed three times for 5 min each in PBS. Slides were incubated with anti-rat IgG Alexa 555 or anti-rabbit IgG Alexa 488 secondary antibody for 1 h and then washed three times for 5 min in PBS. To detect the cell nucleus, slides were mounted using VECTASHIELD with DAPI (Vector Laboratories). The sections were observed and visualized using a fluorescent microscope Leica DM2500 (Leica Microsystems, Wetzlar, Germany). Final image processing and quantification were performed by ImageJ software (NIH).

### 4.17. Statistical Analysis

Data were expressed as mean ± SD of the mean or as mean ± SEM when specified. Student’s t test was used for paired samples for intragroup comparisons and for unpaired samples in the comparison between groups. When the distribution was not normal, the Mann–Whitney U test was used. The Kruskal–Wallis test was used to compare means of more than three groups. Analyses were conducted with GraphPad Prism 8 software (San Diego, CA, USA). Differences were considered statistically significant at *p* < 0.05 with a 95% confidence interval.

## 5. Conclusions

In the present study, we developed an improved MSC cell line that secretes highly immunosuppressive EVs. Taking advantage of gene editing tools and priming of parental cells with pro-inflammatory cytokines, we circumvented some of the main limitations of EV-based therapeutic strategies. Thus, MSC-T-HIF can be used as a source of EVs for longer periods, and there is no necessity to repeat biopsies to obtain MSCs. In addition, the increased potency of EV_MSC-T-HIF_^c^ facilitates the use of EVs in patients, making cell-based therapeutic strategies safer and more robust.

## Figures and Tables

**Figure 1 ijms-22-03416-f001:**
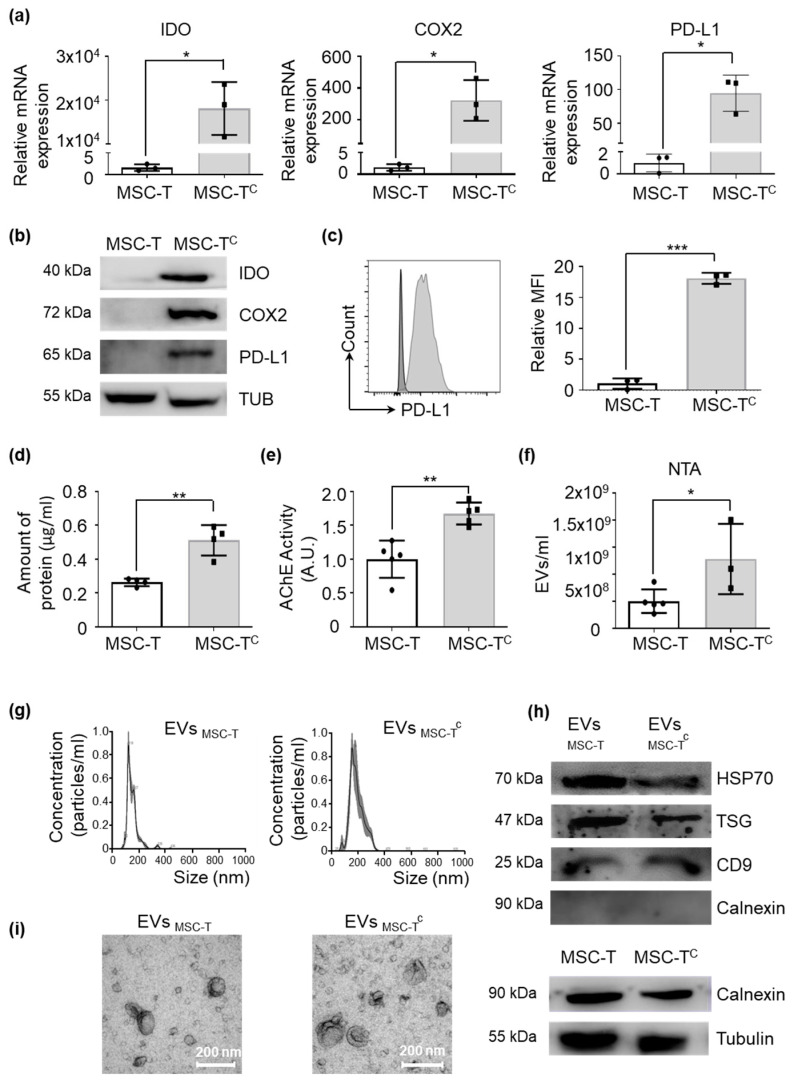
A conditioning medium primes the immunomodulatory capacity of mesenchymal stromal cells (MSCs). Quantification of immunosuppressive molecules and abundance of extracellular vesicles (EVs) released by telomerase-expressing MSCs conditioned (MSC-T^c^) or not (MSC-T) with pro-inflammatory cytokines for 48 h. (**a**) indoleamine 2,3 dioxygenase (*IDO*), cyclooxygenase 2 (*COX2*) and *PD-L1* gene expression levels measured by RT-qPCR. Target gene expression was normalized to *GAPDH* expression and is relative to levels in MSC-T, arbitrarily set to 1. Graphs represent mean ± SD of 3 independent experiments. Paired *t*-test was used for statistics. (**b**) Representative Western blots of IDO, COX2 and PD-L1 protein expression; α-tubulin was used as a protein loading control. (**c**) PD-L1 protein expression measured by flow cytometry using mean fluorescence intensity (MFI) of human telomerase enzyme (MSC-T) (black) and MSC-Tc (grey). Values are represented as relative to MSC-T geometric mean value. Graphs represent mean ± SD of three independent experiments. Paired *t*-test was used for statistics. (**d**) Concentration of protein of EVs extracted from 100 mL of culture medium from 1 × 10^7^ cells that were resuspended in 100 μL of PBS. Graphs represent mean ± SD of four independent experiments. Paired *t*-test was used for statistics. (**e**) Quantification of acetylcholinesterase activity in EVs. Values are represented as relative to the EVMSC-T value. Graphs represent mean ± SD of three independent experiments. Paired *t*-test was used for statistics. (**f**) Amount of EVs per milliliter of supernatant of the same number of MSC-T and MSC-Tc measured by nanoparticle tracking analysis (NTA). Graphs represent mean ± SD of three independent experiments. Unpaired *t*-test was used for statistics. * *p* < 0.05, ** *p* < 0.01. (**g**) Representative images of EVs analyzed by NTA. (**h**) Representative Western blot of Hsp70, TSG101 and CD9 proteins in EVs. Absence of calnexin demonstrates a pure EV preparation. Cells were used as calnexin-positive controls. (**i**) Representative electron microscopy images of isolated EVs collected from different MSC conditions. Scale bar: 200 nm.

**Figure 2 ijms-22-03416-f002:**
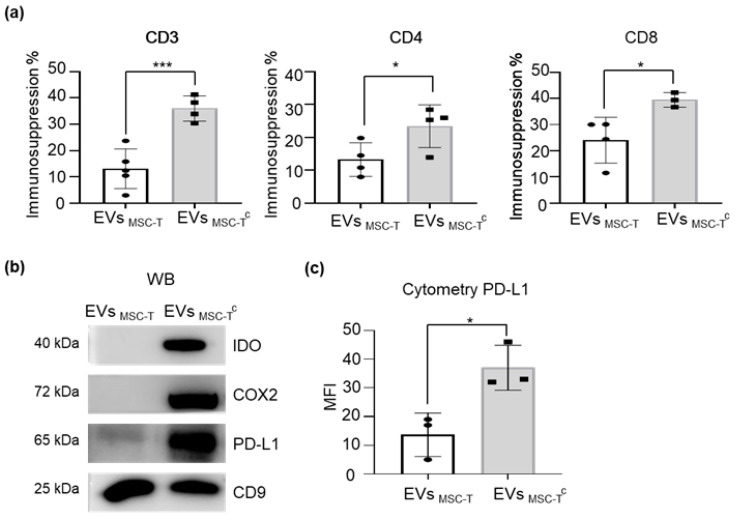
Conditioning results in an enhancement of the immunomodulatory capacity of EVMSC-T. (**a**) Peripheral blood mononuclear cells (PBMCs) were stained with CFSE and stimulated with anti-CD3 and anti-CD28 beads in the presence or absence of EVMSC-T or MSC-T^c^-derived EVs (EV_MSC-T_^c^). After 5 days, cells were stained with anti-CD3, anti-CD4 and anti-CD8 antibodies and proliferation of T-cell subsets was determined by flow cytometry measurement of CFSE dilution. Suppression (percentage) was calculated based on the expansion index. Graphs represent mean ± SD of four independent experiments. Unpaired *t*-test was used for statistics. (**b**) Representative Western blot of IDO, COX2 and PD-L1 protein expression levels. CD9 was used as a loading control. (**c**) PD-L1 protein expression level measured by flow cytometry using MFI geometric mean value of EV_MSC-T_ and EV_MSC-T_^c^ stained with anti-PD-L1. Values are represented as relative to EVMSC-T geometric mean value. Graphs represent mean ± SD of three independent experiments. Paired *t*-test was used for statistics. * *p* < 0.05, *** *p* < 0.001.

**Figure 3 ijms-22-03416-f003:**
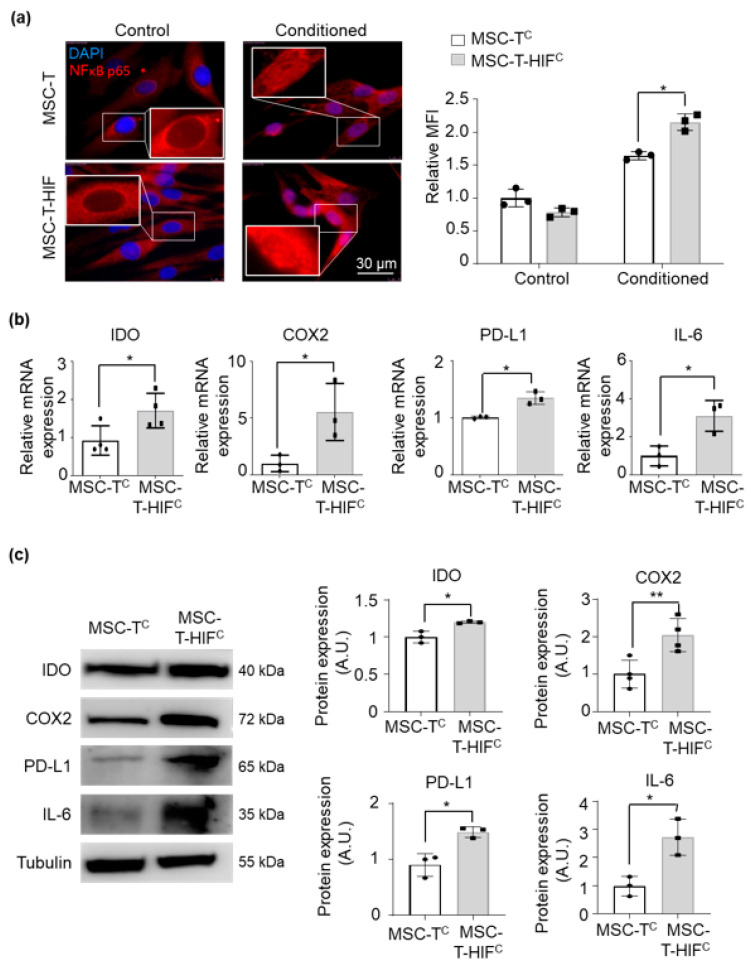
HIF overexpression strengthens signaling through the NF-κB pathway and consequently the expression of immunosuppressive cytokines. (**a**) Immunolocalization of p65 in MSC-T, MSC-Tc, MSC-T with an HIF-1α-GFP lentiviral vector (MSC-T-HIF) and conditioned MSC-T-HIF (MSC-T-HIFc). Red: p65, blue: DAPI. Scale bar: 30 μm. Mean of fluorescence intensity was measured per nucleus and values were normalized to those of MSC-T cells (each point represents the mean of 50 nuclei). Graphs represent mean ± SD of three independent experiments. Sidak’s multiple comparisons test was used for statistics. (**b**) *IL-6*, *IDO*, *COX2* and *PD-L1* expression levels quantified by RT-qPCR in MSC-Tc and MSC-T-HIFc. Expression level of the target gene in each sample was normalized to *GAPDH* expression. Graphs represent mean ± SD of fold change of three independent experiments. Paired *t*-test was used for statistics. (**c**) Representative Western blots of IL-6, IDO, COX2 and PD-L1 proteins. Expression levels were quantified by densitometry relative to MSC-Tc. α-tubulin was used as a protein loading control. Bars represent mean ± SD of three independent experiments. Paired *t*-test was used for statistics. * *p* < 0.05, *p* < 0.01.

**Figure 4 ijms-22-03416-f004:**
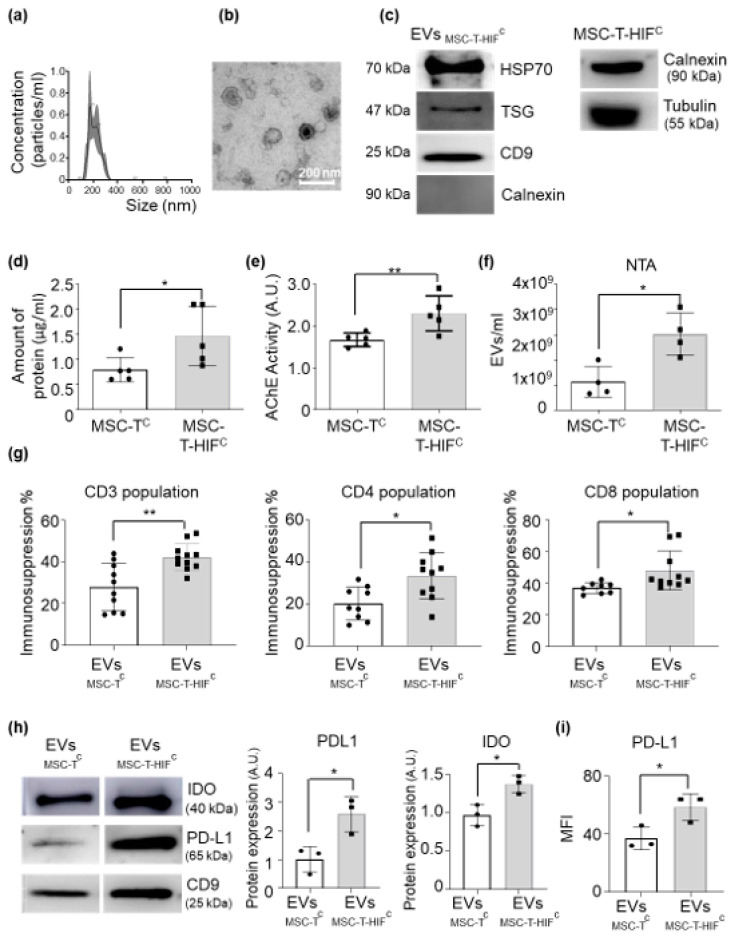
MSC-T-HIF^c^-derived EVs (EV_MSC-T-HIF_^c^) display enhanced immunosuppression properties. (**a**) Representative images of EVs collected from MSC-T-HIFc evaluated by nanoparticle tracking analysis (NTA). (**b**) Representative electron microscopy images of isolated EVs collected from MSC-T-HIFc. Scale bar: 200 nm. (**c**) Representative Western blots of Hsp70, TSG101 and CD9 proteins in EVs. Absence of calnexin demonstrates a pure EV preparation. Cells were used as calnexin-positive controls (**d**) Concentration of protein of EVs extracted from 100 mL of culture medium from 1 × 10^7^ cells that were resuspended in 100 μL of PBS. Graphs represent mean ± SD of six independent experiments. Paired *t*-test was used for statistics. (**e**) Quantification of acetylcholinesterase activity in EV_MSC-T_^c^ and EV_MSC-T-HIF_^c^. Values are represented as relative to the EV_MSC-T_^c^ value. Graph represents mean ± SD of three independent experiments. Paired *t*-test was used for statistics. (**f**) Amount of EVs per milliliter in the supernatant of the same number of MSC-Tc and MSC-T-HIFc measured by NTA. Graphs represent mean ± SD of three independent experiments. Unpaired *t*-test was used for statistics. (**g**) Peripheral blood mononuclear cells (PBMCs) were stained with CFSE and stimulated with anti-CD3 and anti-CD28 beads in the presence or absence of EV_MSC-T_^c^ or EV_MSC-T-HIF_^c^. After 6 days, cells were stained with anti-CD3, anti-CD4 and anti-CD8 antibodies and the proliferation of T-cell subsets was determined by flow cytometry of CFSE dilution. Suppression (percentage) was calculated based on the expansion index. Graph represents mean ± SD of seven independent experiments. Unpaired *t*-test was used for statistics. (**h**) Representative Western blots of IDO and PD-L1 proteins in EV_MSC-T_^c^ and EV_MSC-T-HIF_^c^. Expression levels were quantified by densitometry relative to EV_MSC-T_^c^. CD9 was used as a loading control. Graph represents mean ± SD of three independent experiments. Paired *t*-test was used for statistics. (**i**) PD-L1 protein expression level measured by flow cytometry using the mean fluorescence intensity of EV_MSC-T_^c^ and EV_MSC-T-HIF_^c^. Values are represented as relative to the EV_MSC-T_^c^ geometric mean value. Graphs represent mean ± SD of three independent experiments. Paired *t*-test was used for statistics. * *p* < 0.05, ** *p* < 0.01.

**Figure 5 ijms-22-03416-f005:**
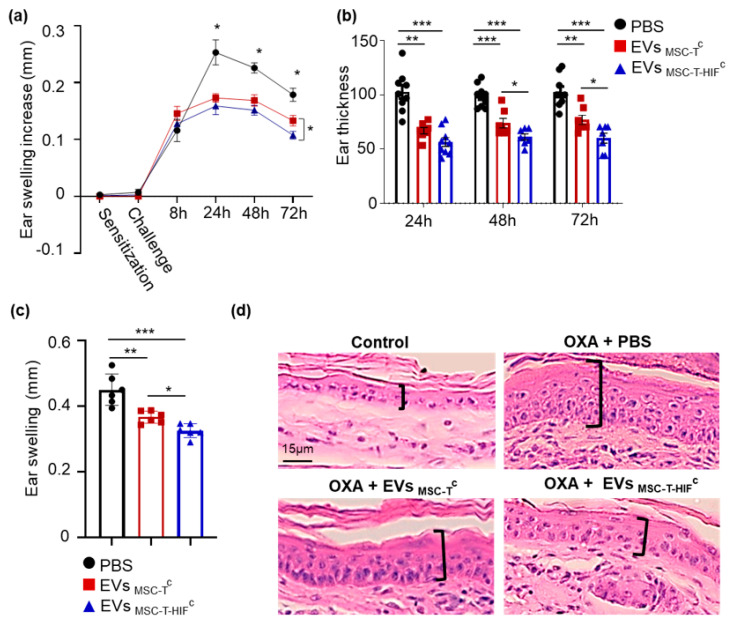
EV_MSC-T-HIF_^c^ show immunoregulatory potential in a delayed-type hypersensitivity mouse model. BALB/c mice were sensitized by applying a 3% oxazolone solution on the abdominal skin at day −5 and the left ear was challenged with 1% oxazolone solution at day 0. Mice were injected with the following different treatments: PBS (black circle), EV_MSC-T_^c^ (red triangle) or EV_MSC-T-HIF_^c^ (blue square) 6 h after challenge. The non-challenged right ear was used as a no-inflammation control. (**a**) Ear swelling was recorded as an increase in thickness (in mm) at different time points. Data are presented as the mean ± SEM of seven mice in each group. Tukey’s multiple comparisons test was used for statistics. (**b**) Ear thickness measured at different time points. Data reflect the reduction in thickness caused by each treatment relative to the PBS control ear. Graph represents mean ± SD of seven mice in each group. Tukey’s multiple comparisons test was used for statistics. (**c**) Ear swelling measurement on histological sections. Bars represent the mean ± SD of thickness of six animals. Paired *t*-test was used for statistics. * *p* < 0.05, ** *p* < 0.01, *** *p* < 0.001. (**d**) Representative histological hematoxylin and eosin-stained sections of ears at the 72 h time point. Epithelial enlargement is noted in brackets. Scale bar: 15 μm.

**Figure 6 ijms-22-03416-f006:**
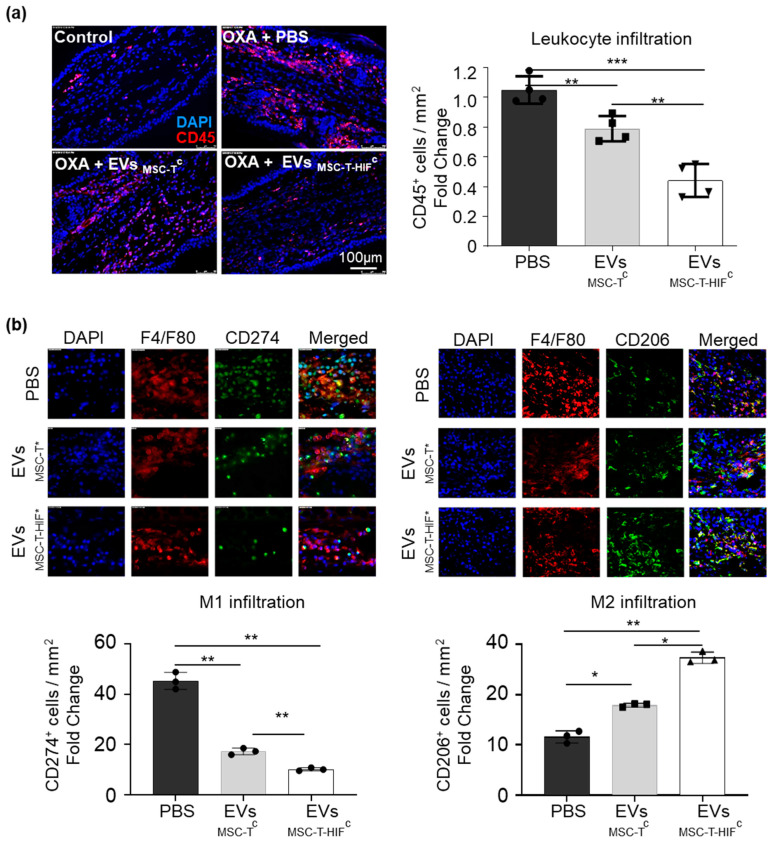
EV_MSC-T-HIF_^c^ reduce leucocyte infiltration and the presence of immunomodulatory (M2) macrophages in a delayed-type hypersensitivity (DTH) mouse model. (**a**) Immunodetection of CD45+ cells in ears at the 72 h time point. Scale bar: 100 μm. Quantification of CD45+ positive cells per mm^2^. (**b**) Immunodetection of F4/F80 (pan-macrophage marker; red) and CD274+ cells (pro-inflammatory (M1) macrophage; green) and CD206 (M2 macrophage; green) in ears at the 72 h time point. Scale bar: 100 μm. Quantification of double positive cells per mm^2^. Data are relative to the PBS condition. Graphs represent mean ± SD of four mice. Paired *t*-test was used for statistics. * *p* < 0.05, ** *p* < 0.01, *** *p* < 0.001.

## Data Availability

All data generated and/or analyzed during this study are included in this published article and its additional files. All the data can be share upon request by email.

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
