# Peer review of "HIF-1α and Pro-Inflammatory Signaling Improves the Immunomodulatory Activity of MSC-Derived Extracellular Vesicles"

_ijms, 2021, doi:10.3390/ijms22073416_

Round 1
Reviewer 1 Report
In their paper the authors set out to show how the immunosuppressive properties of MSC-derived extracellular vesicles can be improved by preconditioning culture medium and HIF-1alpha overexpression.
Major comments:
Figure 1b: Why is it that MSC-T do not show any IDO, COX2, PD-L1 protein expression? Could the authors please discuss?
Sup. Fig. 1e and Sup. Fig. 2e:
The figure legend (Sup. Fig. 1e) states that "doubling time" was analyzed. This does not seem to be correct, because in the formula (is it correct or is one bracket is missing?) "time" is not considered. Please describe correctly.
Please repeat the experiments three times so that statistics could be performed.
In addition, it seems that Sup Fig. 2e is not an independent experiment from Sup Fig. 1e, because the data for MSC and MSC-T seem to be identical. This should be clear to the reader, because otherwise the impression is given that the experiment performed several times in an astonishingly reproducible way.
Sup. Fig. 2 a: Which antibiotic was used for selection?
Sup. Fig. 2c: The figure legend states that the experiment was performed three times. However, only two data points are visible. Experiment should be repeated and statistics should be performed.
Sup. Fig. 2d: Are HIF-1a target genes turned on in normoxia in the MSC-HIF and MSC-T HIF cells? Overall control cells that overexpress GFP alone would have been desirable.
Fig. 3a:
How many cells were included in each experiment?
To make it easier for the reader to see, it would be good to zoom in on a nucleus and show the p65 fluorescence without DAPI overlay.
Fig. 5a, b: Is it possible to show exemplary pictures (maybe in the supplement)?
Discussion, Lines 340 – 343: “In the present study we confirm a previous report [13]….” Quote 13 refers to a review. This makes it difficult for the reader to immediately understand exactly which result could be reproduced.
General: In many figures (e.g. Fig. 1a; Fig. 3a, b, c; Fig. 4h; Fig. 5 e, Sup. 2c, , all of the control values are set equal to 1 (instead of setting the mean of the controls equal to one). Standard deviations of the control values are not shown.
It is not clear which statistical test was used in which experiment.
The generation of MSC-T-HIF cells has been described by the group before (Ref. 19). Please mention this more clearly and discuss differences/similarities to the previous study. E.g. in the previous work proliferation of both CD4+ and CD8+ T cells was dramatically reduced, regardless of HIF-1 alpha overexpression by MSCs. If this is due to conditioning medium an experiment comparing the medium could be performed.
Minor comments:
Please check spelling of hypoxia-inducible factor-1alpha.
Fig. 1g: When printed the font size of the axcis labeling is too small.
Fig. 1 d, e and Fig. 3 d, e: It is hard to compare the results as only arbitrary units are shown.
The discussion focusses on the therapeutic potential of MSCs. A focus on the presented results would be desirable.
Author Response
Dear reviewer,
find enclosed the answer to your comments and suggestions.
Sincerely
Pilar Sepúlveda, PhD

Reviewer 2 Report
In the manuscript by Gómez-Ferrer M et al., the authors have developed an improved MSC cell line using the strategies involved overexpression of human telomerase enzyme and HIF-1α. In addition, they showed that EVs derived from improved MSC can carry immunosuppressive properties and exhibit the immunoregulatory potential in a delayed-type hypersensitivity mouse model. Overall, the manuscript is written nicely and supported by substantial data. Some minor concerns exist and need to be addressed:
- Line 30-38: It is not clear from the background that why this study is needed. Here, a compelling statement can be provided with a clear hypothesis or statement of purpose of this study.
- There are several errors or typos in the manuscript that need to be fixed. Some of them are hygromycin (line 42), immunosuppression (line 158), tetraspanins (line 264), and IFN-y (line 413,417, 424).
- Fig. 1h, 4a: Values on the y- and x-axis are not visible.
- Fig. 1f and h: Compared to unmodified MSC, MSC-Tc released a higher number of EVs. whereas EV marker such as CD9 does not seem to be affected, but in fact expression of TSG101 appeared to be decreased. Any explanation for that?
- Provide molecular weight of the proteins in the western blots.
- EVs released are also comparatively higher in MSC-T-HIFc. So, the increased presence of immunosuppressive molecules is due to the higher number of EVs. Can authors show whether the loading capacity of EVs is also modulated in an improved MSC cell line?
- Fig. 4h. IDO western blot is missing in the figure. Did the authors also investigate the levels of COX-2 in EVMSC-Tc and EVMSC-T-HIFc?
- Although the authors showed the EVs’ carrying potential of immunosuppressive molecules that could be involved in the immunosuppression of CD4 and CD8 subpopulation, they do not provide the direct involvement of those molecules behind immunosuppression. Can authors show the direct involvement of IDO, COX-2, and PD-L1 using the gene knockdown approach?
- Description of Fig. 5e is missing in the result section.
- Fig. 5e: Here authors used CD45 as a common antigen marker to show leukocyte infiltration at the injury site. Can authors also show the specific leukocyte such as neutrophil infiltration using Ly6G? Also, it would be interesting to see whether EVs derived from MSC-Tc and MSC-T-HIFc can modulate the levels of chemoattractant and chemotactic proteins.
- Line 340: Provide the original research article here (reference 13).
Author Response

(The authors gave the same response as above.)

Round 2
Reviewer 1 Report
My questions were answered to my satisfaction.
Author Response
Dear Reviewer,
thank you for your reply. We are pleased to have been able to answer your comments and suggestions.
Sincerely
PIlar Sepúlveda, PhD
Reviewer 2 Report
The authors did extensive work towards the improvement of the work and the points in my original critique have largely been addressed. Here I have suggestions that authors may address during the Proof Stage:
Line 30-38: The newly added line seems confusing to me and I suggest rewriting.
Fig. 4h: Authors can mention in the result description that observation regarding the COX-2 result and therefore “data not shown”.
Author Response
Dear Reviewer,
thank you for your comments. We have modified the abstract to better explain our aim (lines 38-41) in this study and added a comment in results section regarding the expression of COX2 in EVs (lines 287-289).
In addition, the reference to the method for calculating population doubling is number 46 and it has been amended in the methods section (line 554)
We hope that this modifications are satisfactory
Sincerely
Pilar Sepúlveda, PhD